# Subnational tuberculosis burden estimation for Pakistan

**Alvaro Schwalb** [1,2,3]*, **Zia Samad**[4], **Aashifa Yaqoob**[4], **Razia Fatima**[4], **Rein M. G. J. Houben**[1,2]

1 TB Modelling Group, TB Centre, London School of Hygiene and Tropical Medicine, London, United Kingdom, 2 Department of Infectious Disease Epidemiology, London School of Hygiene and Tropical Medicine, London, United Kingdom, 3 Instituto de Medicina Tropical Alexander von Humboldt, Universidad Peruana Cayetano Heredia, Lima, Peru, 4 Common Management Unit for AIDS, TB & Malaria, Ministry of National Health Services, Regulations & Coordination, Islamabad, Pakistan

* alvaro.schwalb@lshtm.ac.uk

**Data Availability Statement:** Data and analysis code are available on GitHub (https://github.com/aschwalbc/SUBsET-PAK).

**Funding:** This work was supported by the Global Fund to Fight AIDS, Tuberculosis and Malaria [PO

## Abstract

Global tuberculosis (TB) burden estimates are aggregated at the national level, despite the likelihood of uneven distribution across and within regions in the same country. Subnational estimates are crucial to producing informed policies and informing budget allocation at more granular levels. In collaboration with the National TB Programme (NTP), we applied a simple and transparent tool to estimate the subnational TB burden in Pakistan. We tailored the SUBnational Burden Estimation for TB (SUBsET) tool to account for the district-level hierarchy of Pakistan. Districts were assigned weighted scores based on population size, level of urbanisation, households with one room, and food insecurity levels. Using the 2022 national TB incidence estimate, we first allocated the burden across administrative units based on data from the 2010–11 TB prevalence survey and subsequently refined this distribution to reflect weighted scores specific to each district. The estimated TB incidence was compared with pulmonary TB notifications to calculate the case detection rate (CDR) for each district. Utilising the updated SUBsET model, we assigned weight scores to 150 districts spanning seven provinces/regions in Pakistan. The estimated TB incidence varied significantly, ranging from 110 (95%CI: 80–145) to 462 (95%CI: 337–607) per 100,000 inhabitants per year. The provinces bearing the highest burden was Sindh (292; 95%CI: 213–384), followed by Khyber Pakhtunkhwa (269; 95%CI: 196–354) and Punjab (243; 95%CI: 177–320). The CDR was below 70% in three-quarters of the districts and over-reporting (>100%) was observed in 10 districts, primarily within Punjab, which suggests that individuals with TB may be crossing district lines to access care. The application of the SUBsET tool through active collaboration with the NTP revealed high heterogeneity in subnational TB incidence in Pakistan, urging a more granular and tailored approach to TB prevention and care. This approach ensured transparency and acceptance of the findings for wider in-country dissemination.

2023-002151 to AS and RMGJH] and the European Research Council [grant number 757699 to AS and RMGJH]. The funders had no role in the study design, collection, analysis, or interpretation of data, writing the manuscript, or the decision to submit the paper for publication.

**Competing interests:** The authors have declared that no competing interests exist.

## Background

Heterogeneity stands as a key characteristic of tuberculosis (TB) epidemiology, displaying distinct geographical differences that are apparent even when comparing regions within the same country [1]. Despite this, current burden estimation is aggregated at the national level [2]. Furthermore, for countries that have conducted nationwide TB prevalence surveys, estimates are commonly reported at the regional level and thus do not delve deeper into the granularity of TB burden at lower administrative units [3, 4]. The physical environment can congregate risk factors which are shared among a population, resulting in high-incidence hotspots [1]. It is of high interest for policymakers to identify high TB burden locations and target TB prevention and care efforts accordingly.

According to the World Health Organization (WHO), the estimated incidence of TB in Pakistan in 2022 was 264 per 100,000 inhabitants (95%CI: 192–347), ranking fifth globally in terms of the highest absolute TB occurrences [2]. A TB prevalence survey conducted in Pakistan in 2010–11 showed some contrast in estimates across the main administrative units [5]. However, as noted above, the TB burden in Pakistan is also likely distributed unevenly within provinces, with TB policy decisions being highly decentralised [5]. There is increasing demand for estimates at the subnational level to inform policies and optimise resource allocation [6]. A hackathon was commissioned by the Pakistan National TB Programme (NTP) to estimate the subnational TB burden [6]. The models used varied approaches, including binomial logistic regression, small area estimation and latent Markov modelling, and self-organising maps, all providing district-level TB prevalence estimates [6]. Although these state-of-the-art modelling approaches produced valuable outputs for planning, their 'black box' nature may make the methods less transparent, potentially hindering the ability to generate ownership and reproducibility by the NTP.

To address these transparency and usability issues, the SUBsET (SUBnational Burden Estimation for TB) tool was developed [7]. Initially implemented in Indonesia and Tanzania, SUBsET operates on the principle of simplicity to promote transparency, thereby improving the uptake and application of results [7, 8]. It uses a widely available and well-known software, facilitating its distribution and usage, and encourages active participation from the NTP for both initial implementation and subsequent use [7, 8]. To help inform future National Strategic Plans and funding requests, the NTP in Pakistan requested the application of the SUBsET tool to provide subnational estimates. In this study, we describe the implementation and present the results of using the SUBsET tool to estimate the subnational TB burden in Pakistan.

## Methods

### Setting and granularity

Pakistan is the fifth most populous country in the world, with over 240 million individuals [9]. Its territory is divided into four provinces (Balochistan, Khyber Pakhtunkhwa, Punjab and Sindh) and three territories (Azad Jammu & Kashmir, Gilgit-Baltistan, and Islamabad Capital Territory) [9]. In collaboration with the NTP, it was decided that subnational estimates for TB should be provided at the district-level. Across these seven administrative divisions, a granularity of 150 districts was adjusted to align with TB notification reporting by the NTP (**Fig 1 and S1 Text**). Subnational administrative boundaries shapefiles were obtained from the *World Food Programme SDI* (**Supplementary Material**) [10].

### Model description

The SUBsET tool runs on Microsoft Excel (Microsoft Corporation, Redmond, WA, USA), which is user-friendly and widely available, facilitating its distribution and usage [7, 8]. In

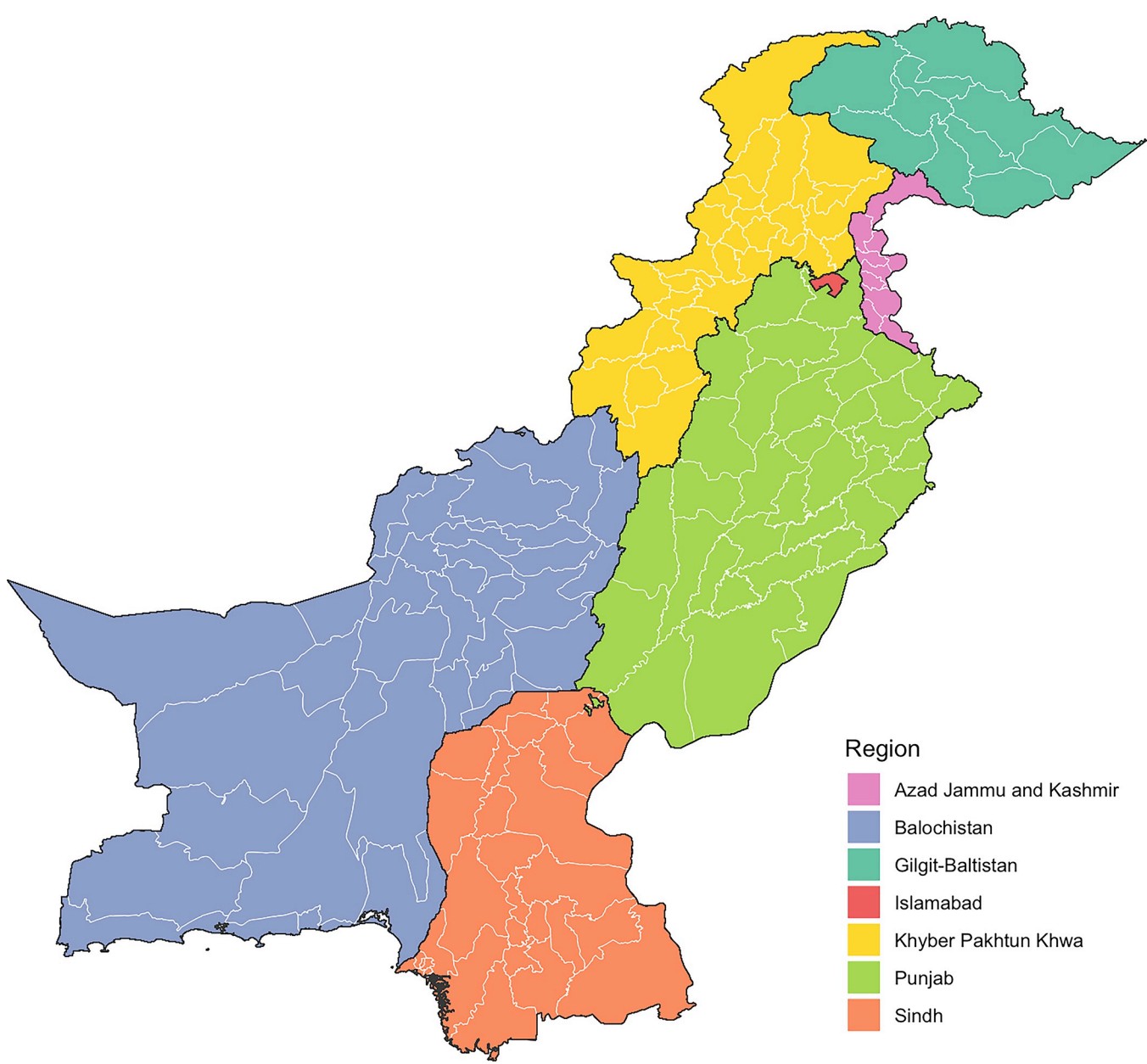

**Fig 1. Administrative and district divisions of Pakistan.** *Subnational administrative boundaries for Pakistan from the World Food Programme SDI under Creative Commons Attribution for Intergovernmental Organisations (CC BY-IGO)* [10].

addition, the tool encourages direct input from the NTP for variable selection, assumptions, and other model-related choices. Detailed methods for the SUBsET tool, including specific formulae for each step, have been previously described by Parwati et al. [7]. Briefly, the tool generates subnational estimates by following these steps:

1. **Determining regional incidence:** The national TB incidence is distributed to each administrative division based on the burden distribution from the TB prevalence survey.

2. **Selecting variables:** Risk factors associated with TB burden are identified, with the expectation that measures of these factors will be available for most (>90%) districts.

3. **Calculating variable weight:** Weights for each variable in a district are calculated by multiplying the district's proportion of the variable by its corresponding relative risk.

4. **Calculating district weight:** The weight for each district is determined by multiplying all its variable weights by its population size.

5. **Distributing TB burden:** The weight scores for each district are summed to derive a total weight score per administrative division. The ratio of each district's score to its administrative-division score is calculated and then multiplied by the regional TB incidence to estimate the TB incidence at the district level.

Additionally, the tool can also be used to estimate the case detection rate (CDR) per district by comparing the district TB burden with the reported TB notifications provided by the NTP. This allows the assessment of healthcare system performance and cross-district health utilisation. CDR categories were set, based on the WHO 70% target [11]: severe underreporting (<25%), underreporting (≥25 to <70%), adequate reporting (≥70 to 100%), and overreporting (>100%).

## Model variables

We used the national TB incidence estimate of 611,000 from the 2022 Global TB Report by the WHO [12]. For subnational distribution across regions, we used the adjusted prevalence rate for bacteriologically confirmed TB from the 2010–11 national TB prevalence survey [5]. There were region-specific TB prevalences available for all except Islamabad Capital Territory as no cluster was randomised in the survey and instead, the average value across regions was taken [5]. Pulmonary TB notification (regardless of bacteriological status) data and population sizes for 2022 were provided by the NTP (**Table 1**).

After consulting with the NTP, three variables were selected due to their association with TB burden and the availability of district-level data: i) proportion of the population living in urban or rural settings, ii) distribution of households by the number of rooms as a proxy for overcrowding, and iii) moderate to severe food insecurity as a proxy for undernutrition [13, 14]. However, complete data for these variables were not available for the regions of Azad Jammu & Kashmir, Gilgit-Baltistan, and Islamabad Capital Territory. For districts with missing data, regional averages were used (**S2 Text**). Then, relative risks (RRs) for each variable were added (**Table 2**); region-specific RRs were available for the level of urbanisation for most regions. Additionally, as part of an exploratory analysis of healthcare system utilisation, the number of TB clinics per district was also documented.

## Analysis

The model was established using Microsoft Excel (Microsoft Corporation, Redmond, WA, USA). It incorporated district-level data, national and regional TB burden estimates, and

**Table 1. Variable sources and definitions.**

| Variable | Year | Definition | Range | Source |
|---|---|---|---|---|
| TB notifications | 2022 | Individuals with TB reported per district | 6–18,445 | NTP |
| Population | 2022 | Number of individuals per district | 57k - 12,898k | NTP |
| Level of urbanisation | 2017–18 | Proportion of population living in urban areas (%) | 0–100 | PDHS [13] |
| Number of rooms | 2019–20 | Proportion of households with one room (%) | 0–72 | PSLM [14] |
| Food insecurity | 2019–20 | Proportion of the population living with moderate and severe food insecurity (%) | 4–49 | PSLM [14] |

NTP: National TB Programme; PDHS: Pakistan Demographic and Health Survey; PSLM: Pakistan Social and Living Standards Measurement Survey.

relative risks (RRs) of selected variables to project the distribution of TB burden. Descriptive analysis and the construction of plots were performed using the R programming language for statistical computing and graphics [16].

### Sensitivity analysis

To further understand the relative importance of the selected variables, we compared the district-level output of our comprehensive model with a secondary model that distributed the burden based solely on population size.

### Ethical considerations

Ethics approval was not necessary since all data used were either publicly available or anonymised at the time of analysis. Permission for access to district-level TB notification data was granted by the NTP. The implementation and revision of the model were conducted in conjunction with the NTP, incorporating direct input from regional managers.

## Results

### TB incidence

The distribution of TB burden across Pakistan was highly heterogeneous, with incidence rates ranging from 110 cases (95%CI: 80–145) per 100,000 inhabitants in Harnai, Balochistan, to 462 cases (95%CI: 337–607) per 100,000 inhabitants in Khairpur, Sindh (**Fig 2** and **S1 Table**). Compared to the estimated national TB incidence of 264 per 100,000 inhabitants, 56 districts had a higher estimated incidence (**S1 Table**). The Sindh region accounted for the majority of Pakistan's TB burden, with an incidence rate of 292 (95%CI: 213–384) per 100,000 inhabitants. This was followed by Khyber Pakhtunkhwa and Punjab, with incidence rates of 269 (95%CI: 196–354) and 243 (95%CI: 177–320) per 100,000 inhabitants, respectively (**Table 3**).

### CDR

Regionally, only Punjab and Gilgit-Baltistan recorded a CDR above 70% (**Table 3**). Across three-quarters of Pakistan's districts, the CDR was below 70%, with over-reporting observed in 10 districts (**Fig 3** and **S1 Table**). The CDR varied significantly, ranging from as low as 6% in Kharmang (Gilgit-Baltistan) and Kalat (Balochistan) to a high of 288% in Diamir (Gilgit-Baltistan). Notably, in Gilgit-Baltistan, districts with severe underreporting are situated near those overreporting.

**Table 2. Relative risk of selected variables.**

| Variable | Region | Relative risk (95%CI) | Source |
|---|---|---|---|
| Level of urbanisation[α] | Azad Jammu & Kashmir | 0.95 (0.24–3.67) | Pakistan TB Prevalence Survey 2010–11 [5] |
| | Balochistan | 0.65 (0.47–0.90) | |
| | Gilgit-Baltistan | 0.65 (0.47–0.90) | |
| | Islamabad | 0.65 (0.47–0.90) | |
| | Khyber Pakhtunkhwa | 0.39 (0.18–0.87) | |
| | Punjab | 0.72 (0.54–0.97) | |
| | Sindh | 0.48 (0.31–0.77) | |
| Number of rooms[β] | All regions | 2.00 (1.00–2.50) | SUBsET tool application in Indonesia [7] |
| Food insecurity[γ] | All regions | 3.20 (3.10–3.30) | Population attributable factor for undernutrition [15] |

[α]Proportion of the population living in urban areas. [β]Proportion of households with one room
[γ]Proportion of the population living with moderate and severe food insecurity.

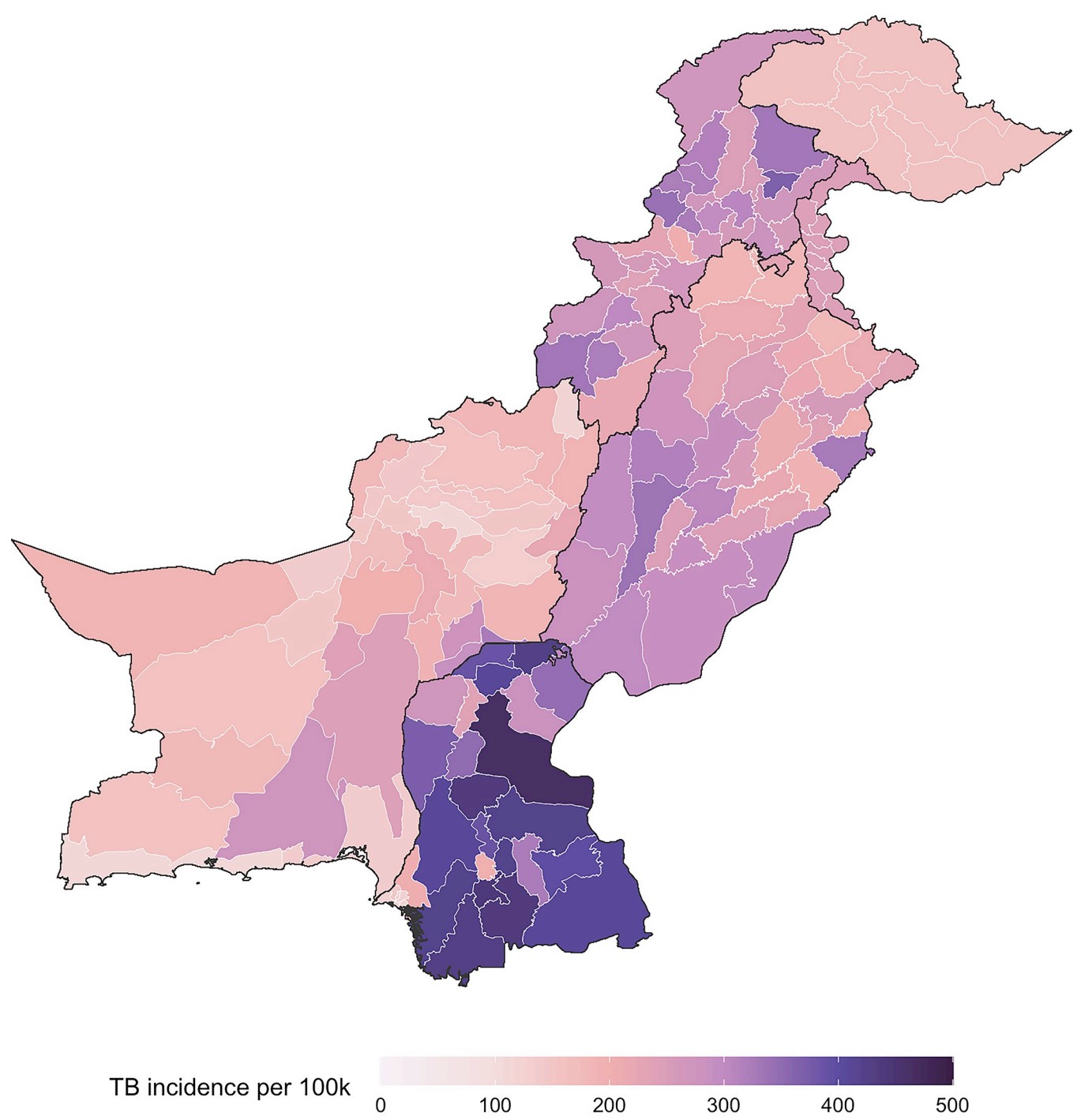

TB incidence per 100k

0    100    200    300    400    500

**Fig 2. District-level TB incidence in Pakistan.** *Subnational administrative boundaries for Pakistan from the World Food Programme SDI under Creative Commons Attribution for Intergovernmental Organisations (CC BY-IGO)* [10].

## Exploratory analysis

Scatterplots were constructed to evaluate the correlation between TB incidence and CDR with the number of healthcare facilities per 100,000 inhabitants. There was a weak negative correlation (R = -0.17; p = 0.038) between TB incidence and the number of healthcare facilities but no

**Table 3. Regional estimates of TB burden and case detection rate in Pakistan.**

| Region | TB incidence per 100k (95%CI) | Case detection rate (95%CI) |
|---|---|---|
| Azad Jammu & Kashmir | 243 (177–320) | 43% (33–59%) |
| Balochistan | 178 (130–235) | 38% (29–52%) |
| Gilgit-Baltistan | 162 (118–213) | 96% (73–132%) |
| Islamabad | 230 (168–303) | 26% (20–36%) |
| Khyber Pakhtunkhwa | 269 (196–354) | 32% (24–44%) |
| Punjab | 243 (177–320) | 75% (57–102%) |
| Sindh | 292 (213–384) | 57% (43–78%) |

correlation (R = -0.03; p = 0.723) between the CDR and the number of healthcare facilities per 100,000 inhabitants (**Fig 4**).

## Sensitivity analysis

Heterogeneity in TB incidence estimates is introduced with the addition of selected variables (**Fig 5**). We found five districts with a higher estimated incidence and eight with a lower estimated incidence when using the comprehensive model compared with the secondary model that distributed the burden based solely on population size.

## Discussion

The TB burden in Pakistan exhibits considerable heterogeneity, with estimated incidences ranging from 110 to 462 per 100,000 inhabitants annually. The development of the model was greatly enriched by contributions from the NTP, facilitating a thorough exploration of the TB burden across 150 districts. This implementation demonstrates that the SUBsET tool remains an intuitive and adaptable solution for subnational TB burden estimation, easily tailored to the administrative divisions of a country.

Identifying districts with high TB incidence is crucial for the effective allocation of resources, especially when there is a significant disparity compared to neighbouring districts. For this, the demand for TB modelling is increasing as available data can be harnessed to inform programmatic action [17, 18]. Additionally, previous modelling exercises have indicated that prioritising TB prevention and care efforts in a localised hotspot could have an impact on TB incidence comparable to targeting broader community areas [19]. This suggests that guiding local policies and targeted interventions in these high-incidence areas could be a strategic approach in the fight against TB.

As mentioned before, these are not the first subnational estimates generated for Pakistan, and even for the same data sources, the geographical distribution of TB varied considerably across models [6]. While a formal comparison with the model outputs described in Alba et al. is beyond the scope of this paper, we subscribe to the main conclusion that substantial differences are likely to exist [6]. Unfortunately, as with the other efforts, there is limited ability to validate estimates against empirical data. While notifications provide some insight into the distribution at the district level, given the many potential distorting layers between burden and notifications, we chose not to use notifications in the formulation of our estimates. Instead, we used them to get an understanding of the performance gap and healthcare-seeking behaviour.

The CDR analysis highlights several low-performing districts situated near those with over-reporting, suggesting that individuals with TB may be crossing district lines to access care. Similarly, the models developed during the TB Hackathon consistently identified districts with low notification-to-prevalence ratios [6]. Foremost, this could reflect a mismatch between

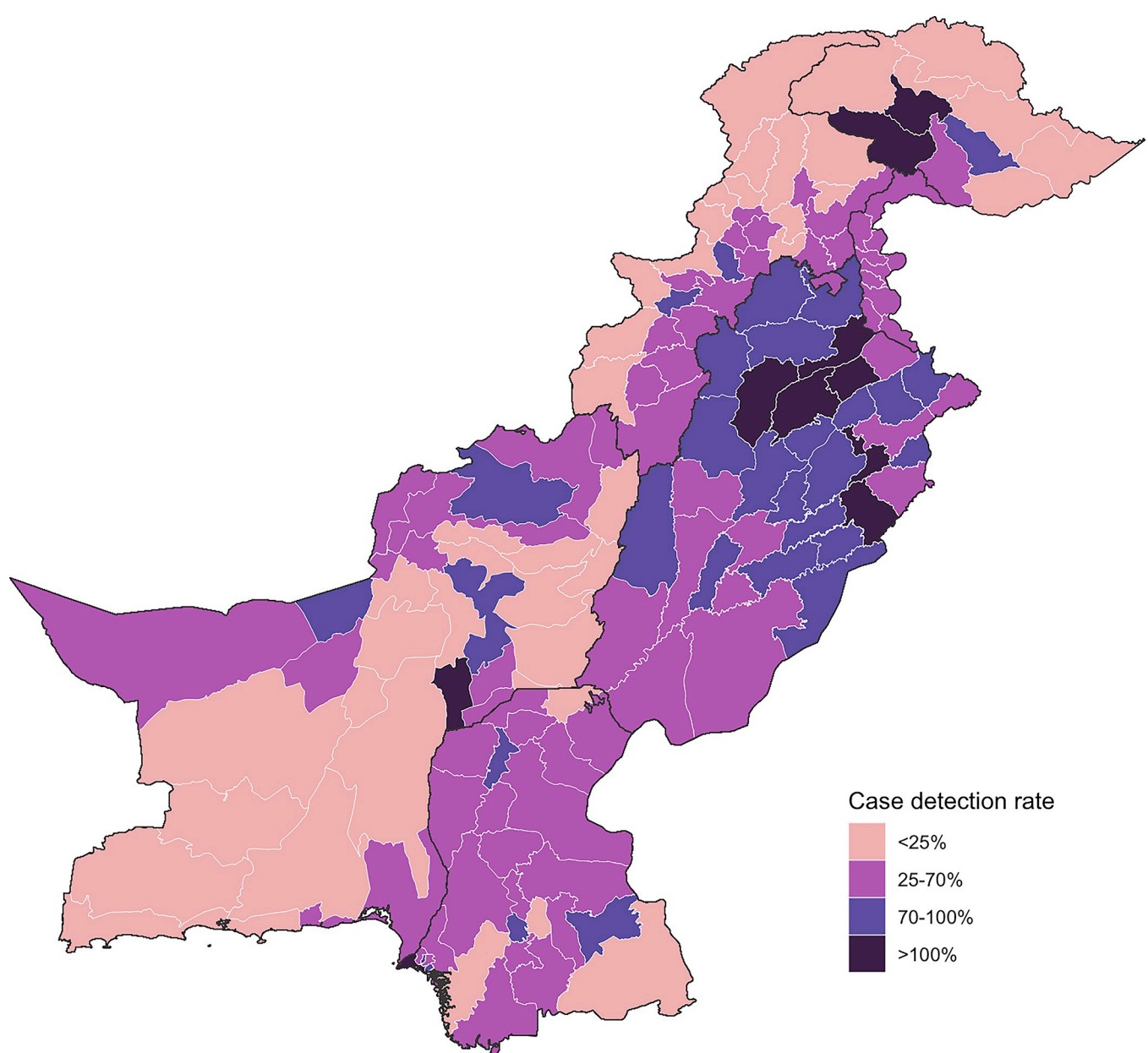

**Fig 3. District-level case detection rate in Pakistan.** *Subnational administrative boundaries for Pakistan from the World Food Programme SDI under Creative Commons Attribution for Intergovernmental Organisations (CC BY-IGO)* [10].

empirical notification data and model estimates that emerge based on methods and assumptions made. The CDR outputs from our model, although dependent on the accuracy of our estimates, provide a valuable starting point for investigating underperformance issues in these districts. Notably, Pakistan is among the top ten countries with the largest gap between TB notifications and the best estimates of TB incidence [2]. Therefore, to effectively narrow this gap, special emphasis is necessary on districts that are markedly underreporting.

Given the simplicity of the model and the availability of the data, a few limitations should be addressed. Firstly, the regional burden distribution uses a dated TB prevalence survey. While a decline might be expected with an updated survey, regional distribution is unlikely to

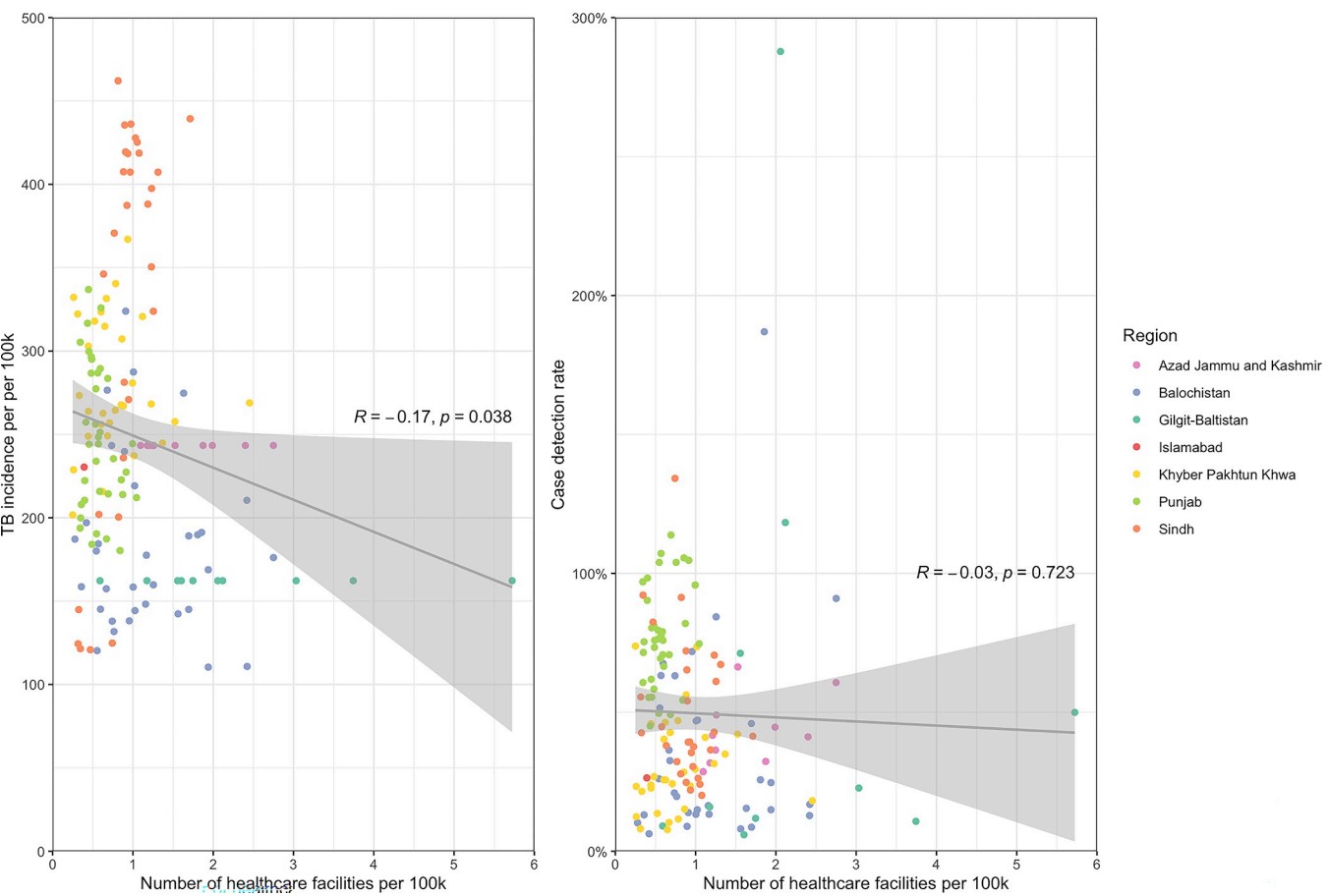

**Fig 4. Correlation between TB incidence and CDR with the number of healthcare facilities in Pakistan.** The Pearson correlation coefficient (R) is a measure of linear correlation between two variables and takes a value between -1 and 1. The p-value reflects the probability of observing a non-zero correlation coefficient in our sample when the null hypothesis assumes that there is no linear correlation.

change significantly due to generational stability of sociodemographic determinants within regions. Furthermore, in contrast to other models, the simplicity of the model allows for future adaptation, making estimates readily available after new surveys are conducted. Secondly, the selection of variables at the district level contributing to TB burden was limited. Key sociodemographic characteristics and underlying conditions such as HIV infection and diabetes mellitus, known to be associated with TB burden [20], lacked the necessary granularity for inclusion in the model. For instance, the absence of certain key variables in Azad Jammu & Kashmir and Gilgit-Baltistan prevented the estimation of a weighted distribution of TB burden at the district level. Therefore, it is advisable to primarily rely on regional estimates in these cases. Future model iterations should incorporate these and other risk factors, if district-level data become available, as they could provide additional depth to the burden estimation.

Additionally, RRs for selected variables were not available at the regional level, except for urbanisation levels derived from TB prevalence survey data [5]. Incorporating region-specific RRs would refine the weight scores used in burden distribution calculations. Our estimates also do not account for the uncertainty of RRs. Although upper and lower bounds are provided, based on the confidence interval of the national TB incidence rate for Pakistan, the bounds of the RRs themselves were not incorporated into the model. Moreover, TB notifications in the model are currently only accounting for pulmonary TB (regardless of

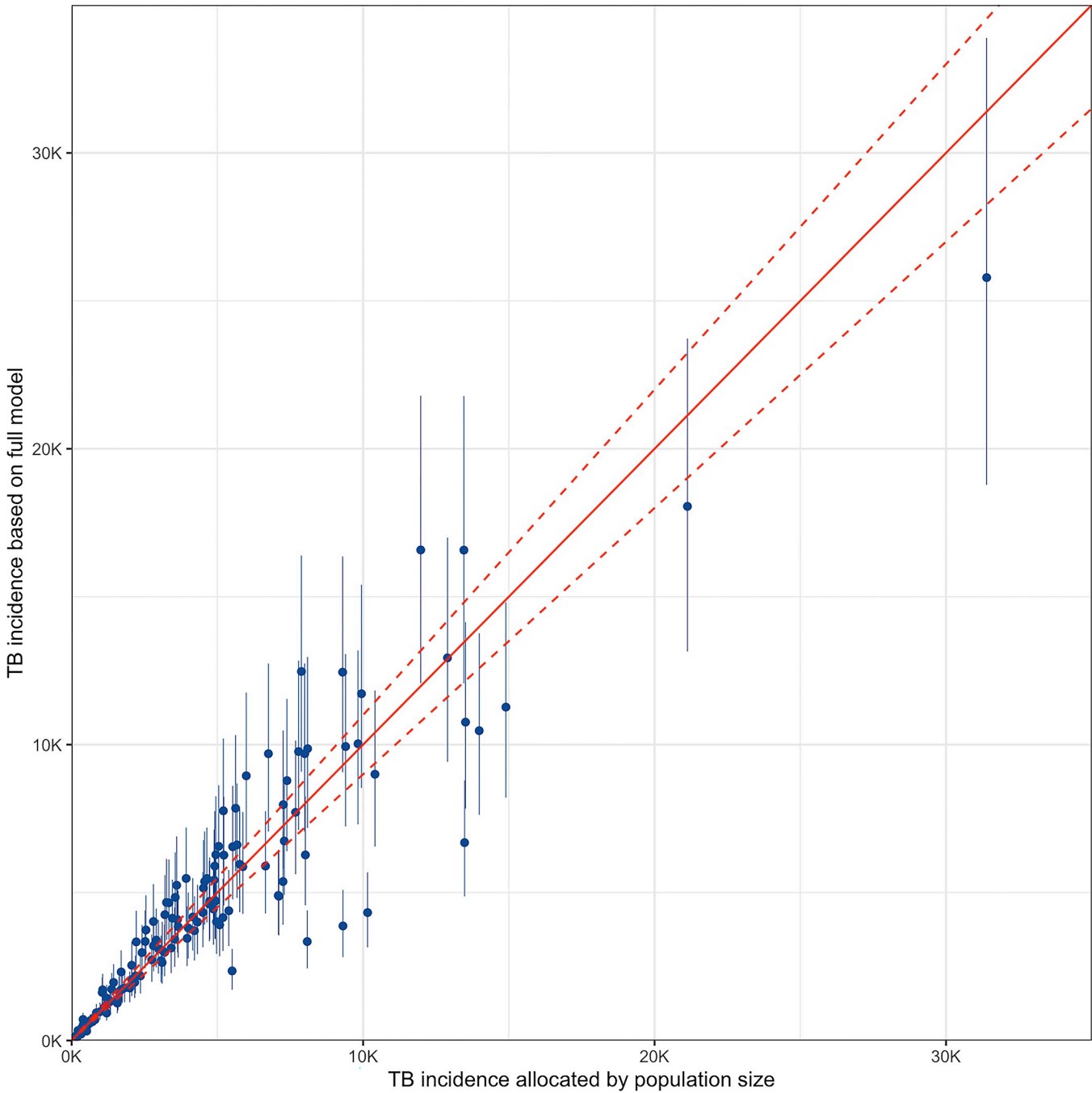

**Fig 5. TB incidence heterogeneity captured by model variables.** Shows the change in estimated absolute incidence per district when comparing the comprehensive model with a secondary model that distributed the burden based solely on population size. Data points above the diagonal line indicate districts with higher estimates in the comprehensive model, while those below the line indicate districts with lower estimates compared to the secondary model.

bacteriological status) instead of all forms of TB (i.e., including extrapulmonary TB), as reported in the Global TB Report. This choice was made to tailor the results to the TB prevalence survey which focused on pulmonary TB estimates. Finally, the exploratory analysis does not consider the size and complexity of healthcare facilities, thus some could manage the number of patients in an area while others might not despite their combined number.

Despite its limitations, the SUBsET tool generates subnational estimates of TB burden through a transparent and accessible approach. Developed in collaboration with the NTP at each stage, the tool is well-informed by both data and policymakers, as with previous iterations. The NTP provides most of the input data and advises on which variables should be included for burden distribution. Additionally, the tool's Excel-based platform makes it accessible to non-modellers by clearly presenting its functionality. The final version of the tool is designed to be dynamic, allowing for updates and improvements as new data become available.

## Conclusion

In this subsequent application of the SUBsET tool, TB burden distribution was estimated through active collaboration with the NTP. Through its simplicity and user-friendly interface, the NTP is able to implement updates as new data becomes available. Overall, this approach ensured both transparency in methodology and acceptance of the findings. The estimates from this modelling study have been endorsed by the provincial and regional managers and are now being utilised for policy planning.

## Supporting information

**S1 Text. District divisions of Pakistan.**
(DOCX)

**S2 Text. Data completeness.**
(DOCX)

**S1 Table. District estimates of TB burden and case detection rate in Pakistan.**
(DOCX)

## Acknowledgments

The authors would like to acknowledge the help of Dr Asad Zaidi for his external support of this project.

## Author Contributions

**Conceptualization:** Alvaro Schwalb, Rein M. G. J. Houben.

**Data curation:** Alvaro Schwalb, Zia Samad, Aashifa Yaqoob.

**Formal analysis:** Alvaro Schwalb.

**Supervision:** Razia Fatima, Rein M. G. J. Houben.

**Writing – original draft:** Alvaro Schwalb.

**Writing – review & editing:** Zia Samad, Aashifa Yaqoob, Razia Fatima, Rein M. G. J. Houben.

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
