## [Decision Letter · Decision Letter 0]

15 Jul 2024

PGPH-D-24-01312

Subnational tuberculosis burden estimation for Pakistan

Dear Dr. Schwalb,

Thank you for submitting your manuscript to PLOS Global Public Health. After careful consideration, we feel that it has merit but does not fully meet PLOS Global Public Health’s publication criteria as it currently stands. Therefore, we invite you to submit a revised version of the manuscript that addresses the points raised during the review process.

We look forward to receiving your revised manuscript.

Kind regards,

Brian Wahl

Academic Editor

Journal Requirements:

Additional Editor Comments (if provided):

Reviewers' comments:

Reviewer's Responses to Questions

**Comments to the Author**

1. Does this manuscript meet PLOS Global Public Health’s publication criteria? Is the manuscript technically sound, and do the data support the conclusions? The manuscript must describe methodologically and ethically rigorous research with conclusions that are appropriately drawn based on the data presented.

Reviewer #1: Yes

Reviewer #2: Yes

2. Has the statistical analysis been performed appropriately and rigorously?

Reviewer #1: Yes

Reviewer #2: Yes

3. Have the authors made all data underlying the findings in their manuscript fully available (please refer to the Data Availability Statement at the start of the manuscript PDF file)?

Reviewer #1: No

Reviewer #2: Yes

4. Is the manuscript presented in an intelligible fashion and written in standard English?

Reviewer #1: Yes

Reviewer #2: Yes

5. Review Comments to the Author

Reviewer #1: This is a well-written article demonstrating the potential of a simple, Excel-based model for identifying regions that might benefit from increased resources for TB detection. Identifying approaches for targeted resource delivery in a heterogeneous TB epidemic is critical to reducing inequities and achieving TB control. However, there are a few methodological limitations, namely the lack of validation and unclear implementation strategy, that I believe need to be addressed.

Line 181 – The authors suggestion that the CDR over 100% suggests patients crossing district lines to access care seems more appropriate for the discussion section than the methods

Line 225 – How do the authors distinguish between under-performance and cross-district health utilization when it comes to districts with low CDRs?

Low CDRs may represent lower than expected notifications (due to undiagnosed TB or cross-district care seeking), but could also reflect overestimates of the true prevalence. Given the limitations of the data used in the model (especially using subnational prevalence data that is over a decade old – as this data may not reflect the current situation), I think the authors should discuss this possibility.

The authors do not discuss the relative importance of the variable selection process. What are the implications, and how might the results vary with different variables or different RRs? A sensitivity analysis may be warranted here.

The authors clearly describe the use of the SUBsET tool and the resulting output from the model. The accessibility and ease of use of an Excel-based tool is convincing. However, the authors do not appear to discuss what this implementation might look like; given the strong collaboration with the NTP, it would be nice to see more specific ideas for how this tool might be used, beyond simply running the model and targeting resources to high-burden or underreporting areas, which can be done with any model.

Data availability is a key limitation, and this is well-noted by the authors. However, it is difficult the see the utility of this, or any model, without reliable and timely data. Can the authors please clarify how they envision the use of this model in a programmatic setting, given that relying on old prevalence surveys may be providing out-of-date information?

I think it would be helpful to have more discussion comparing the findings of other subnational estimates – where are the discrepancies noted? Are there certain regions or contexts where estimates seem the most reliable (or unreliable)? Do different models produce consistently higher estimates than other? What other differences between model outputs are notable? While I understand the challenges in formal validation of these model outputs, it would be helpful to have a more specific understanding of how they compare in order for readers to determine whether this model is more or less appropriate than others.

Reviewer #2: The topic is very interesting and the manuscript is well-written. For reproducibility of research, a Github link is provided for data and relevant materials. Moreover, the manuscript can be accepted for publication.

6. PLOS authors have the option to publish the peer review history of their article (what does this mean?). If published, this will include your full peer review and any attached files.

**Do you want your identity to be public for this peer review?** For information about this choice, including consent withdrawal, please see our Privacy Policy.

Reviewer #1: No

Reviewer #2: No

---

## [Decision Letter · Decision Letter 1]

29 Aug 2024

Subnational tuberculosis burden estimation for Pakistan

PGPH-D-24-01312R1

Dear Dr. Schwalb,

We are pleased to inform you that your manuscript 'Subnational tuberculosis burden estimation for Pakistan' has been provisionally accepted for publication in PLOS Global Public Health.

Best regards,

Brian Wahl

Academic Editor

Reviewer Comments (if any, and for reference):

Reviewer's Responses to Questions

**Comments to the Author**

1. If the authors have adequately addressed your comments raised in a previous round of review and you feel that this manuscript is now acceptable for publication, you may indicate that here to bypass the “Comments to the Author” section, enter your conflict of interest statement in the “Confidential to Editor” section, and submit your "Accept" recommendation.

Reviewer #1: All comments have been addressed

2. Does this manuscript meet PLOS Global Public Health’s publication criteria? Is the manuscript technically sound, and do the data support the conclusions? The manuscript must describe methodologically and ethically rigorous research with conclusions that are appropriately drawn based on the data presented.

Reviewer #1: Yes

3. Has the statistical analysis been performed appropriately and rigorously?

Reviewer #1: Yes

4. Have the authors made all data underlying the findings in their manuscript fully available (please refer to the Data Availability Statement at the start of the manuscript PDF file)?

Reviewer #1: Yes

5. Is the manuscript presented in an intelligible fashion and written in standard English?

Reviewer #1: Yes

6. Review Comments to the Author

Reviewer #1: All my comments have been addressed. Thank you to the authors for your thorough and thoughtful responses.

7. PLOS authors have the option to publish the peer review history of their article (what does this mean?). If published, this will include your full peer review and any attached files.

**Do you want your identity to be public for this peer review?** For information about this choice, including consent withdrawal, please see our Privacy Policy.

Reviewer #1: **Yes: **Katherine Robsky
